

# Transport of biodeposits and benthic footprint around an oyster farm, Damariscotta Estuary, Maine

Kara Gadeken, William C. Clemo, Will Ballentine, Steven L. Dykstra, Mai Fung, Alexis Hagemeyer, Kelly M. Dorgan and Brian Dzwonkowski

Dauphin Island Sea Lab, Dauphin Island, AL, United States of America
Department of Marine Sciences, University of South Alabama, Mobile, AL, United States of America

## ABSTRACT

The benthic impact of aquaculture waste depends on the area and extent of waste accumulation on the sediment surface below and around the farm. In this study we investigated the effect of flow on biodeposit transport and initial deposition by calculating a rough aquaculture "footprint" around an oyster aquaculture farm in the Damariscotta River, ME. We also compared a site under the farm to a downstream "away" site calculated to be within the footprint of the farm. We found similar sediment biogeochemical fluxes, geochemical properties and macrofaunal communities at the site under the farm and the away site, as well as low organic enrichment at both sites, indicating that biodeposition in this environment likely does not have a major influence on the benthos. To predict accumulation of biodeposits, we measured sediment erodibility under a range of shear stresses and found slightly higher erosion rates at the farm than at the away site. A microalgal mat was observed at the sediment surface in many sediment cores. Partial failure of the microalgal mat was observed at high shear velocity, suggesting that the mat may fail and surface sediment erode at shear velocities comparable to or greater than those calculated from *in situ* flow measurements. However, this study took place during neap tide, and it is likely that peak bottom velocities during spring tides are high enough to periodically "clear" under-farm sediment of recent deposits.

# INTRODUCTION

As demand increases to feed the world's growing population, many are looking toward dramatically expanding marine aquaculture as an efficient and sustainable means of food production (*Kobayashi et al., 2015*; *Gentry et al., 2017*). Growth of the aquaculture industry seems to be rising to meet that demand; worldwide animal aquaculture production increased from less than 20 million tons in the early 1990s to ~80 million tons in 2016 and is rapidly approaching parity with rates of wild seafood capture (*FAO, 2018*). As aquaculture production increases and more marine aquaculture operations open in coastal areas,

Corresponding author
Kara Gadeken, kgadeken@disl.org

the need for an understanding of the local impacts of aquaculture grows more pressing, particularly of the under-farm sediment environments receiving sinking aquaculture waste.

Past reports of the local benthic impact of aquaculture range from dramatic alteration of bottom characteristics to apparently few effects depending on the cultured organism, the method of culture and local environmental conditions (*Kaiser et al., 1998*; *Newell, 2004*; *Crawford, Macleod & Mitchell, 2003*; *Forrest et al., 2009*). The impacts of aquaculture are commonly assessed by comparing sediment characteristics and communities directly under an aquaculture farm to a less impacted or reference site (*Grant et al., 1995*; *Findlay & Watling, 1997*; *Mallet, Carver & Landry, 2006*; *Higgins et al., 2013* and others). However, the deposition of aquaculture biodeposits, and therefore their eventual impact, depends on the physical drivers that transport the biodeposits to the benthos. Direct measurements of particle sinking rate and horizontal distance traveled are lacking in the literature (*Callier et al., 2006*), although several studies have numerically modeled them (*Dowd, 2005*; *Navas, Telfer & Ross, 2011*; *Comeau et al., 2014*; *Silva et al., 2019*). Also sparse are descriptions of post-deposition sediment dynamics; depending on the local flow environment, settling particles that reach the benthos may remain there or be removed by erosion.

The footprint of an aquaculture farm is determined by the intensity of lateral flow and settling rate of the particles. In areas with low flow, biodeposits would be expected to accumulate under the farm, increasing the organic content and driving high influxes of oxygen and effluxes of nutrients as organic matter is degraded (*Forrest et al., 2009*). Higher flow rates may be expected to increase the transport distance and therefore the overall size of the farm footprint, likely resulting in a gradient of impact with distance from the farm. However, very high flow may erode under-farm sediments and transport biodeposits a considerable distance from the farm, smearing the edges of the footprint or clearing it completely. Thus the "footprint" does not have a discrete value, but rather needs to be defined explicitly from the range of particle sizes and flow velocities in the system. In tidal systems, erosion may only occur periodically during ebb and flood tides when flow velocities peak or even episodically during peak spring tides. If sediments are resuspended or fluidized, they are exposed to more oxygen, increasing remineralization rates and depleting their labile organic content compared to surface sediments (*Aller, 2004*). Thus, erosion and redeposition of biodeposits may reduce the effect of organic enrichment even more than predicted by transport alone. This study aims to characterize the footprint around an aquaculture farm, specifically addressing the questions: (Q1) How far are biodeposit particles expected to travel from an aquaculture farm before deposition? (Q2) Once deposits have settled to the benthos, do they remain there or are they eroded away?

Impacts of aquaculture can be assessed from differences in the sediment environment under the farm compared with outside of the footprint. Sediments within the footprint are expected to be finer grained and more organic-rich because biodeposits tend to be comprised of finer grained materials than bulk deposited sediment (*Haven & Morales-Alamo, 1966*). Nutrient enrichment from biodeposits may also encourage growth of a microalgal or microbial mat, increasing sediment surface chlorophyll *a* (chl-a), binding surface sediments and altering nutrient and oxygen fluxes (*Mirto et al., 2000*;

*Giles, Pilditch & Bell, 2006*; *Walker & Grant, 2009*). When deposition of biodeposits under an oyster farm results in substantial organic enrichment of sediments, the composition of the macrofaunal community may be altered, though this may vary depending on the local environment of the farm (*Forrest et al., 2009*). Organic enrichment and physical disturbance both tend to reduce the body size, abundance, and diversity of sediment macrofauna (*Pearson & Rosenberg, 1978*; *Aller & Stupakoff, 1996*). Sediments with high organic enrichment tend to host macrofaunal communities comprising hypoxia- and sulfide-tolerant taxa (*Pearson & Rosenberg, 1978*), whereas physical disturbances like sediment erosion, resuspension, and resettlement favor highly mobile, near-surface-dwelling, deposit-feeding taxa (*Brenchley, 1981*; *Aller & Stupakoff, 1996*). Differences in the functional traits and life histories of macrofauna may suggest the kind of disturbance under-farm sediments experience most frequently.

The Damariscotta River Estuary, Maine, is an ideal system for this study, as it experiences strong tidal currents (*Garrett, 1972*; *Brooks, 2009*). Mean flow rates throughout a tidal cycle in a narrow section of the Damariscotta are ~1.1 m s$^{-1}$, with peak velocities far higher (*McAlice, 1977*). Aquaculture in Maine estuaries is also quickly growing; the total value of Maine oyster aquaculture rose from less than $1 million in 2005 to nearly $9.7 million in 2019 (State of Maine Department of Marine Resources, https://www.maine.gov/dmr/aquaculture/data/index.html). Much of that increase is attributable to expansion of aquaculture in the Damariscotta River, which alone constitutes approximately two-thirds of the total oyster production in Maine and increased in harvest yield from ~1.3 million to ~9.4 million oysters between 2005 and 2019. Despite the industry's rapid growth, there is little existing research examining the impacts of oyster aquaculture on the local benthos. Our objective was to describe the footprint of an oyster farm in the Damariscotta River, ME, and assess our estimate of the footprint by comparing sediments at sites under the farm and downriver of the farm but within the same depth and flow regime.

## MATERIALS & METHODS

### Study area

The Damariscotta is a narrow, partially mixed mesotidal estuary in the midcoast of Maine. The study was conducted at Perkins Point, approximately 20 km from the mouth of the Damariscotta River (44 00.016′N, 69 32.650′W), where a large oyster aquaculture operation is located on a 24-acre (~97,000 m$^2$) lease (Mook Sea Farm, Fig. 1). Verbal permission to use the site for the study was obtained from Bill Mook, owner/operator of Mook Sea Farm. Eastern oysters (*Crassostrea virginica*) are grown at the farm in mesh bags held in rigid, rectangular floating OysterGro$^{TM}$ cages that are strung together in rows. The farm sits above a sloping shoal with a mean water depth ranging from ~2–4 m. We assessed our estimate of the footprint by comparing a site directly underneath the oyster cages ("Farm" site) in the southern portion of the farm with one downstream, ~90 m south of the edge of the farm lease area ("Away" site). The Away site location was constrained by the geomorphology of the river; because the river bends and narrows
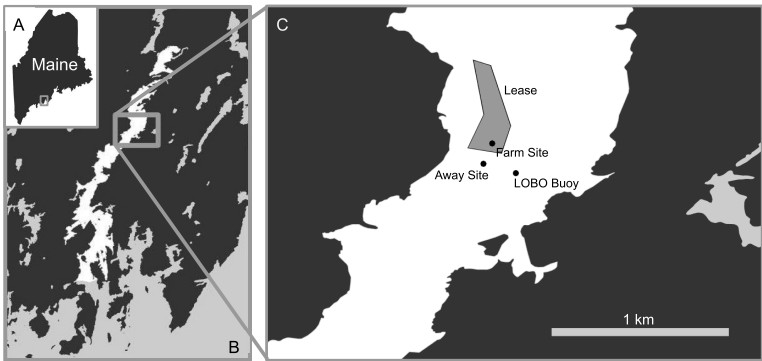

**Figure 1** **Map of study area.** The study took place at the Perkins Point location of Mook Sea Farm (marked "Lease" in C) in the Damariscotta River (white in B) in Maine, USA (A). The farm sits on a sloping shoal, while the surface LOBO buoy is located in the channel.

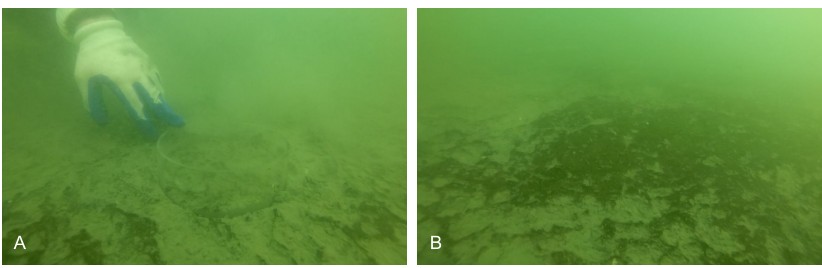

**Figure 2** **Images of benthos from diver core collection.** (A) Acrylic core in sediment with sparse microbial mat. (B) Patchy microbial mat on sediment surface. Photo credit: Christopher Rigaud, University of Maine, Scientific Diving Program.

downstream (Fig. 1), selecting a site farther than ∼90 m would risk it being in an area too hydrodynamically different from the Farm site for comparison. Instruments were deployed on moorings at both sites to measure hydrographic properties, with sediment traps to quantify sedimentation. Sediment cores were also collected at each site for evaluation of sediment erodibility, geochemical characteristics, biogeochemical fluxes, and macrofaunal community structure. While diving to collect the cores for this study, we observed benthic microalgal mats in distinct patches of varying sizes (Fig. 2), though we did not discriminate between areas with and without mats in our sampling.

## Flow and deposition

To determine the horizontal distance particles would be transported before deposition (Q1), flow velocity near the bottom was measured over a tidal cycle with an Acoustic Doppler Velocimeter (ADV) and combined with laboratory measurements of particle settling velocities. Here we define the footprint based on the settling velocity of the mean size of biodeposits and the maximum distance transported over the tidal cycle during neap tide. Because mean water column height varied from ∼2–4 m across the farm, calculations were done with the maximum and minimum depths to provide a range. Thus, biodeposit

particles smaller than the mean would travel beyond our defined footprint under peak flow velocities, but the footprint would encompass most of the tidal range in deposition. To demonstrate the effect of particle size on the footprint, we also calculated transport distance for the 5th and 95th percentile of particle sizes.

The ADV was horizontally mounted to a metal tripod mooring deployed at the Farm site, measuring velocity (8 Hz) 37 cm above the bottom. An Onset HOBO dissolved oxygen (DO) logger was also mounted to the mooring (50 cm above the bottom) to measure near-bottom DO to determine whether low oxygen occurred that might affect biogeochemical fluxes or infauna. A cinder block mooring was deployed at the Away site. Temperature and salinity measurements were collected near-surface and at bottom at both sites to measure stratification. Water level at the farm was measured by the pressure sensor on the ADV. Moorings were deployed for ∼23 h on August 2–3, capturing nearly two tidal cycles during a neap tidal period. Because the experiment was conducted over a short time frame, hydrographic data for the entire summer season was obtained from a Land/Ocean Biogeochemical Observatory (LOBO) surface mooring ∼250 m southeast of the farm (part of the Sustainable Ecological Aquaculture Buoy Network; http://maine.loboviz.com).

Sediment traps were attached to the bottom moorings at both sites to measure sedimentation rates. The traps were made of 5.08 cm inner diameter PVC cylinders, 15 cm in length, and capped on one end, following *Gardner (1980)*. Traps on the bottom mooring at the Farm site were diver-deployed and intended to measure a combination of biodeposits directly from the farm as well as ambient deposition. The Away site traps were attached to the cinder block mooring that was lowered to the bottom from the boat and were intended to measure the ambient particle deposition without influence from the farm. The mooring was repositioned once it was on the bottom which may have created a sediment plume that falsely inflated sedimentation data at that site, so those data should be interpreted with caution. Additional traps were attached directly to the underside of the oyster cages at the Farm site for a direct estimate of *in-situ* particle production by the oysters without under-farm flow transporting the particles. Comparison of these data to deposition at the bottom mooring took into account the area of the farm occupied by gear (∼40% - obtained from the farm's lease application). Six traps were deployed during two 24-hr deployment periods separated by 5 days, during which the tidal range varied by ∼0.5 m. All traps were deployed and retrieved at high tide. At least one trap was lost from each location during each deployment. During the first period, traps were deployed and recovered from the Farm site ($n = 4$) and the cages ($n = 5$). During the second period, sediment traps were deployed and recovered from the Away site ($n = 5$) and the cages ($n = 5$). After collection, sediments were dried for 24 h at 60 °C and weighed.

To measure biodeposit settling velocities, biodeposits were collected from adult, harvest-size oysters held in a flow-through seawater system. Biodeposits were spread in a petri dish to prevent particle aggregation, maintaining the size and cylindrical shape of freshly released biodeposits observed in both the lab and field. Deposits were then pipetted into the middle of a 37.8 L aquarium of seawater, at least 12 cm away from any wall to prevent wall effects. Videos of their descent were recorded using a Nikon D5300 DSLR camera equipped with an AF-S Micro NIKKOR 105 macro lens. It was not possible to distinguish between
feces and pseudofeces from the observations, and we did not describe the composition of the biodeposit particles. The descents of 100 individual particles were recorded. Each particle's dimensions (length and diameter of cylindrical particle) were measured in ImageJ and the angle (normal to vertical) and rate of descent tracked with the ImageJ MtrackJ plugin (*Schneider, Rasband & Eliceiri, 2012*; *Meijering, Dzyubachyk & Smal, 2012*). Settling velocity depends on density and particle surface area. The length and width of each particle measured from the videos were used to calculate surface area using the geometric formula for cylinder surface area. We were unable to measure the densities of individual pellets, but density was calculated using a form of Stokes' settling equation specifically modified for small cylinders (*Komar, 1980*),

$$V_t = 0.079 \frac{1}{\mu} \left( \rho_p - \rho_w \right) g L^2 \left( \frac{L}{D} \right)^{-1.664} \tag{1}$$

where $V_t$ is settling velocity, $\mu$ is the viscosity of the water, $\rho_p$ and $\rho_w$ are the densities of the particle and water respectively, $g$ is acceleration due to gravity, L is particle length, and D is particle diameter. Particle surface area and angle of descent were compared to settling velocity using a multiple regression. Density was not included in this model, as it was calculated from the settling velocity.

An estimate of biodeposit tidal transport ($L_f$) was calculated as:

$$L_f = \frac{u_s(t) h(t)}{w_s} \tag{2}$$

using the bottom horizontal velocity ($u_s$) measured with the ADV and the mean particle settling velocity ($w_s$) calculated from the lab observations. The height of the water column ($h(t)$), or vertical distance traveled, and bottom horizontal velocity ($u_s(t)$) changed depending on the time ($t$) in the tidal cycle. To provide an upper and lower bound on the potential biodeposit tidal transport, calculations used the deepest and shallowest mean water depths at the farm (2 and 4 m). It should be noted that this calculation uses velocity measured with the ADV which is assumed to be in the bottom boundary layer, so the transport distances should be viewed as conservative estimates.

## Erosion

To determine whether settled biodeposits were likely to be resuspended (Q2), we calculated bottom shear velocity throughout the flood/ebb tidal cycle and conducted laboratory erosion experiments across a similar range of bottom shear velocities.

We used the flow velocities measured by the ADV to calculate bottom shear velocity throughout the tidal cycle *via* three indirect methods (logarithmic profile, covariance and turbulent kinetic energy methods) as described by *Kim et al. (2000)* for high sampling rates (>5 Hz). Multiple methods are commonly used and compared, as methods can be biased by local conditions such as waves, stratification, and bedforms (*e.g.,* *Sherwood, Lacy & Voulgaris, 2006*; *Pieterse et al., 2015*). The logarithmic profile method assumes a logarithmic velocity profile using the von Karman-Prandtl equation. The reliability of the estimates from this method is severely limited by having only a single point velocity measurement. However, single point techniques have been developed

(*Sherwood, Lacy & Voulgaris, 2006*) and *Kim et al.*'s (*2000*) methodology was followed. The covariance method requires the velocity measurements to be in the constant stress layer of the bottom boundary (a reasonable assumption for the instrument depth and low water column stratification, buoyancy frequency ~0.004) and uses turbulent fluctuations in the along-estuary direction and vertical flow velocities to determine the bottom stress. The turbulent kinetic energy method assumes a linear relationship between turbulent kinetic energy and bottom stress. Bottom shear stress ($\tau_b$) from both of these methods is then linked to shear velocity as $\tau_b = \rho u_*^2$ where $\rho$ is water density and $u_*$ is shear velocity. As seen in other studies, all three methods produced results with similar patterns, but peak magnitudes showed considerable differences (*e.g.*, (*Kim et al., 2000*; *Sherwood, Lacy & Voulgaris, 2006*; *Pieterse et al., 2015*)). Because the covariance method is less affected by local conditions (*e.g.*, stratification, bedforms; (*Sherwood, Lacy & Voulgaris, 2006*)) and the relationship between the along-estuary velocity and shear velocity was well represented by a quadratic fit ($u_* = 0.27096u_s^2 + 0.00089u_s + 0.00466, R = 0.93$), we used that estimate for our further analyses. The effects of waves on bed stress were not considered because of the short fetch of the estuary.

We conducted laboratory erosion experiments using a custom-built Gust erosion chamber to generate near-uniform bed shear (*Gust & Muller, 1997*; *Thomsen & Gust, 2000*; *Tengberg et al., 2004*; U-GEMS Manual, *Green Eyes LLC, 2015*). Triplicate 10 cm diameter sediment cores were collected by divers from the Farm and Away sites and placed in a large holding tank with flowing seawater from the Damariscotta River. All erosion tests were performed within 60 h of collection. Each core was capped with the Gust chamber, and a rotating disc within the chamber generated increasing levels of shear velocity (0.30 cm s$^{-1}$, 0.95 cm s$^{-1}$, 1.34 cm s$^{-1}$, 2.01 cm s$^{-1}$, 2.32 cm s$^{-1}$). Each shear velocity level was maintained for 20 min before increasing disc rotation and flow rates. For each 20 min shear velocity level, water and eroded material were removed by a pump at the center of the disc at 10 cm above the sediment surface and filtered through 47 mm Whatman GF/F filters (1.5 μm pore size). Effluent was replaced with the same seawater supplied to the core holding tank, so suspended mass in the pre-erosion core overlying water and the replacement water was likely similar. The lowest shear velocity, 0.30 cm s$^{-1}$, was used as a flushing step and was not filtered for suspended sediment analysis (no resuspension was observed at this low shear). Filters were then dried at 60 °C for 24 h and weighed to obtain eroded sediment mass for each shear velocity level. One Away core was not used in the analysis due to accidental introduction of a large nereid worm from an unrelated experiment in the holding tank. Suspended sediment concentration, $C_s$ (kg m$^{-3}$), for each site at each velocity level was calculated from the dry mass (kg) of filtered sediment divided by the volume (m$^{-3}$) of water filtered. $C_s$ was converted to eroded mass per area ($E$; kg m$^{-2}$):

$$E = \frac{C_s V_c}{A_c} \tag{3}$$

where $V_c$ is chamber volume ($7.9 \times 10^{-4}$ m$^3$), and $A_c$ is sediment surface area within the core ($7.9 \times 10^{-3}$ m$^2$). Due to small and unequal sample sizes, statistical comparisons between sites and stress levels were not performed.

To generate specific shear velocities, cap rotation and pumping rate were set using calibration equations from the University of Maryland Center of Environmental Science Gust Erosion Microcosm System (U-GEMS) Manual (*Green Eyes LLC, 2015*):

$$u^*_{15} = 0.0318 n^{0.763} \tag{4}$$

$$Q = -28.31 u^{*2}_{15} + 170.2 u^*_{15} - 23.85 \tag{5}$$

where $u^*_{15}$ is shear velocity at 15 °C (cm s$^{-1}$), $n$ is rotations per minute, and $Q$ is pumping rate (mL min$^{-1}$). Shear velocity at 15 °C was converted to shear velocity at the water temperature measured during the erosion tests (22 °C) as:

$$u^*_{15} = u^*_{22}[1 + 0.006(22 - 15)] \tag{6}$$

where $u^*_{22}$ is shear velocity at 22 °C (cm s$^{-1}$) (U-GEMS Manual, *Green Eyes LLC, 2015*). For those more familiar with stress as a measure of shear, the corresponding applied bottom shear stresses ($\tau_b$; Pa) generated by the Gust chamber were 0.009 Pa, 0.09 Pa, 0.18 Pa, 0.41 Pa, and 0.55 Pa. These were calculated from shear velocities ($u^*_{22}$; m s$^{-1}$) as:

$$\tau_b = \rho u^{*2}_{22} \tag{7}$$

(U-GEMS Manual, *Green Eyes LLC, 2015*). A $\rho$ of 1,021 kg m$^{-3}$ was used based on an average temperature of 22 °C and salinity of 31 psu.

This field sampling occurred during neap tide, and also coincided with the highest surface water temperatures of the year when thermal stratification likely peaked. To gain a broader sense of the potential for sediment resuspension during other times in the tidal cycle, near-surface turbidity and velocity data from the LOBO surface mooring east of the farm were analyzed for longer-term patterns. While turbidity is not a direct measure of sediment resuspension, water clarity is typically related to sediment concentrations. Potential relationships among precipitation, water level, current velocity and turbidity were investigated using both the instantaneous data and 40-hour low-pass filtered signals. Near-surface velocity data were also used to estimate bottom shear velocities in higher-flow times in the spring-neap cycle. Near-surface velocity was converted to a proxy for near-bottom velocity based on a scaling factor determined from the relationship between the LOBO buoy and the ADV-measured velocities during the overlap period in August. This near-bottom velocity proxy was used to estimate shear velocity using a quadratic relationship between the longitudinal velocity and direct calculations of the shear velocity (using the covariance method). This fitted relationship was extrapolated to predict shear velocity expected from the highest flow in the spring-neap tidal cycle.

### Geochemical characterization

Sediments at both sites were evaluated for grain size distribution, water content, percent organic content, and concentrations of C and N. One diver-collected 10-cm diam core from each site was vertically sectioned. Water content was calculated as the difference between wet and dry masses after sediment was dried at 60 °C for 24 h. Percent organic content was

calculated as the difference between dry and ash masses after burning sediment at 500 °C for 4 h. Burned sediment was then soaked in a 1% sodium metaphosphate dispersant solution for 24 h to rehydrate and disaggregate sediment prior to grain size analysis. Grain size distribution was determined using a Malvern Mastersizer 3000 particle analyzer (Malvern Panalytical, Malvern, UK). Data were analyzed using Gradistat (Kenneth Pye Associates, Ltd., Berkshire, UK). Preparation of C and N samples included an acidification step to remove inorganic carbon. Concentrated HCl was added to dried sediment samples and heated at 80 °C until the fluid evaporated as described by *EPA (2009)*. Acidified and non-acidified samples were analyzed for C:N to determine the proportion of inorganic carbon. C and N were analyzed on a CHNSO elemental combustion system analyzer (mod. ECS 4010; Costech Analytical Technologies, Valencia, CA). C:N is reported as a molar ratio (mmol g$^{-1}$ sediment).

## Biogeochemical fluxes

To determine whether the "Away" site exhibited similar impacts of oyster biodeposits on biogeochemical fluxes as the site directly under the farm, sediment core incubations were performed in the dark to measure nutrient, DIC and oxygen fluxes and sediment surface samples were collected for chlorophyll a (chl-a) analysis. In an additional treatment, "Amended" cores, we added oyster feces and pseudofeces onto the top of two cores from the "Away" site. Because of the lack of available data for this area on typical respiration rates, the Amended treatment was included to determine whether Away site sediments were being enriched with biodeposits; if sediments from the Away site were receiving large influx of organic matter from the farm, *i.e.,* were saturated, additional organic matter would be expected to have minimal effect on nutrient and oxygen fluxes. The usefulness of the Amended cores therefore depended on "background" (*i.e.,* unenriched) respiration rates being relatively low.

Incubations of sediment cores were performed to measure sediment oxygen flux and nutrient ($NH_4^+$ and $PO_4^{3-}$) and dissolved inorganic carbon (DIC) fluxes. Only dark incubations were performed so that maximum respiration rates could be measured without the influence of microalgal photosynthesis. Our measurements are therefore not representative of daytime *in situ* fluxes. A total of 12 15.2-cm diam sediment cores were taken by divers during two collections (2 and 7 August 2018). During each collection, two cores were taken from the Farm site and four were taken from the Away site. Care was taken during collection that each core had at least 10 cm of sediment depth and at least four cm of overlying water. The cores were brought back to the lab and kept in tanks with flowing water from the Damariscotta River until use in the incubations (<36 h post-collection). Immediately before the start of the incubations, eight mL wet volume of oyster feces and pseudofeces were pipetted onto the top of two cores collected from the Away site and designated the Amended cores. Between the two samplings, this resulted in a total of four replicates each of Farm, Away and Amended cores.

In incubations, the cores were each submerged in separate 5-gallon buckets of seawater and watertight caps affixed to the tops (after methods and Fig 2 in *Dorgan et al., 2020*). The caps allowed for six cm of overlying water and were each equipped with a stir-bar to gently

circulate the overlying water, inflow and outflow taps for water sampling, and an Atlas Scientific DO probe (Atlas Scientific LLC, Long Island City, NY) that measured oxygen concentrations in the overlying water of each core every ∼12 s. The buckets containing the cores were covered with foil to prevent oxygen production by photosynthesis from corrupting oxygen consumption rate measurements. The incubations of cores collected on 2 August provided simultaneous oxygen, nutrient and DIC fluxes. Nutrient and DIC samples were gathered during the incubation of the 7 August cores, however the oxygen measurement instrument failed to record during the initial incubation, so a second incubation was done on the same cores the following day to collect data to calculate oxygen flux. Sediment samples for chl-a were taken from the top ∼1cm of the cores following the second incubation and frozen for later analysis. The samples were prepared using methods adapted from *Welschmeyer (1994)* and analyzed on a Turner Trilogy 7200 using an HCl acidification step to measure chl-a and phaeopigment concentrations.

Nutrient and DIC samples were siphoned from overlying core water at three timepoints using methods described by *Lehrter et al. (2012)*. Samples were taken at the start (0 h elapsed), middle (2 h elapsed) and end (4 h elapsed) of the incubations. DIC samples were kept dark and cold in 20 mL glass vials until processed (<24 h post-collection) on a TOC Carbon analyzer. Nutrient samples were filtered through 25 mm Whatman GF/F filters (0.7 um pore size) and frozen in the dark until processed for $NH_4^+$ and $PO_4^{3-}$ at the Dauphin Island Sea Lab. $NH_4^+$ was measured fluorometrically with modifications as described by *United States Enviromental Protection Agency (2012)* and *Holmes et al. (1999)*. $PO_4^{3-}$ was measured on a spectrophotometer with slight modifications from methods described in *Grasshoff, Ehrhardt & Kremling (1983)*. Rates of change in overlying water concentration for oxygen, DIC and each nutrient were calculated from the linear regression of the respective analyte concentrations *versus* time. These rates of change ($mmol\ min^{-1}$) were multiplied by the height of overlying water to obtain flux rates ($mmol\ m^{-2}\ d^{-1}$). We were unable to obtain oxygen flux values for the Amended cores because they lacked a consistent oxygen trend. Some cores produced several successive oxygen slopes within the same range of oxygen concentrations because low-oxygen water was replaced with high oxygen water during the sampling for nutrients. To calculate a single oxygen flux for each of these cores, time-weighted average slopes were calculated. Shapiro–Wilk and Kruskal–Wallis tests were performed on nutrient and oxygen flux data to determine if there were differences in fluxes between sites (*R Core Team, 2016*).

## Macrofaunal community structure

After data collection, all incubation cores were sieved (500 μm) and the remaining material preserved in 95% ethanol and rose Bengal stain. Macrofauna were picked out of each sample, and individuals identified to family level. Statistical comparison of abundances between the two sites was not performed due to small and uneven sample size. Data from replicate cores were analyzed for Shannon diversity (H') and Pielou's evenness (J'), averaged by site, then Student's t-tests were performed on the diversity and evenness metrics to compare the two sites. An NMDS analysis was performed on square-root transformed

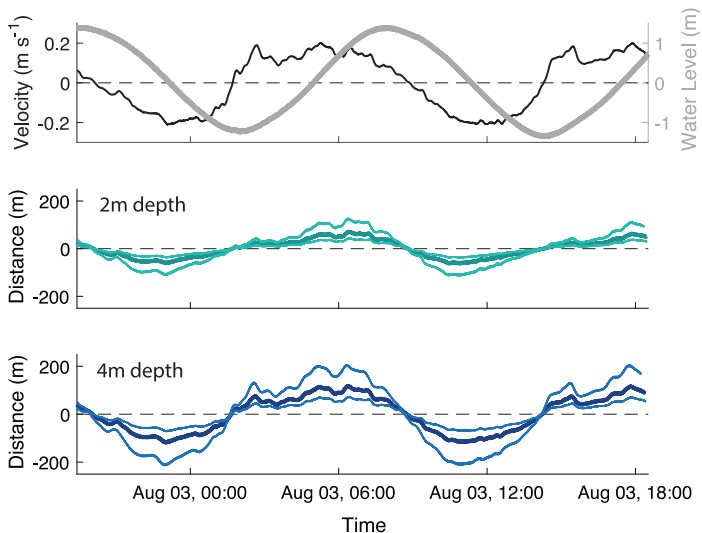

**Figure 3** **Transport of particles throughout the tidal cycle.** (A) Velocity measured 37 cm above the bed (black line, ±20 cm s$^{-1}$) and water level (gray line, 2.86 m range) at the farm site, and calculated distances that biodeposit particles travel in the (B) 2 m depth section and (C) 4 m depth section of the farm over a ~23 h period. In panels B and C, the thick, darker line is the mean and the thin, lighter lines are the 5th and 95th percentile of particle sizes. Positive values indicate flow and transport in a landward direction, while negative indicates a seaward direction.

data (to decrease weighting of highly abundant groups) to compare community structure between sites (PRIMER v7, PRIMER-E Ltd, Plymouth).

## RESULTS

### Flow and deposition

The velocity data were strongly tidal during the deployment, with a clear semi-diurnal cycle. The water level had a 2.86 m range, with a mean water depth of 3.4 m at the away site and 3.5 m at the farm site. The ADV measured near-bottom velocities ranging ± 20 cm s$^{-1}$ (Fig. 3A). While the magnitude of the flood and ebb currents were similar, there was temporal asymmetry in duration, with flood tide being ~1.9 times longer than ebb. Bottom dissolved oxygen remained above 8 mg L$^{-1}$ throughout the sampling period (data not plotted).

The rate of deposition calculated from sediment traps attached to the bottom mooring beneath the farm was (490 ± 36 g m$^{-2}$ d$^{-1}$, $n = 4$). The cage-area normalized rates of deposition calculated from the traps attached to the cages were (163 ± 86 g m$^{-2}$ d$^{-1}$, $n = 5$) for the first deployment and (230 ± 132 g m$^{-2}$ d$^{-1}$, $n = 5$) for the second. These rates were calculated by multiplying the deposition rate calculated from the beneath-cage traps by the percent area of the farm occupied by gear (40%), following the methods of *Testa et al. (2015)*. The highest deposition rate was at the Away site (892 ± 87 g m$^{-2}$ d$^{-1}$, $n = 5$), where deployment and repositioning of the cinder block mooring may have resuspended sediments.

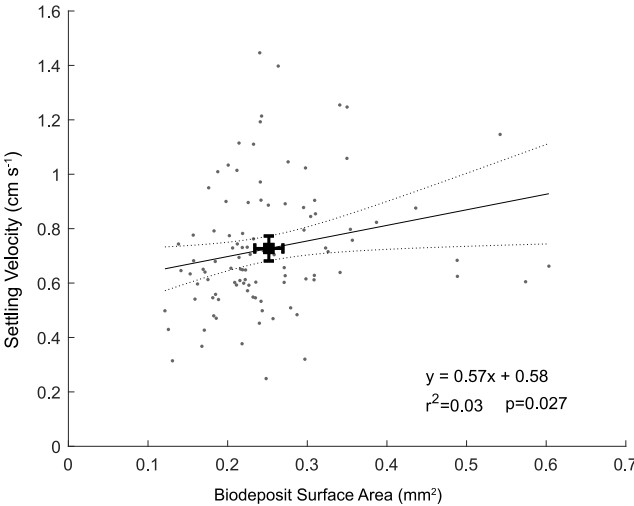

**Figure 4 Particle settling velocity plotted as a function of surface area.** Black square represents means of surface area and settling velocity with associated 95% confidence intervals.

The biodeposits were cylindrical in shape, with an average length of $0.64 \pm 0.14$ mm and average diameter of $0.11 \pm 0.02$ mm. Biodeposit surface area ranged from 0.12 mm$^2$ to 0.60 mm$^2$ with an average surface area of $0.25 \pm 0.09$ mm$^2$. For each one mm$^2$ increase in surface area, we observed a 0.57 ($\pm 0.53$; $\pm$ 95% CI) cm s$^{-1}$ increase in settling velocity ($p = 0.03$; $r^2 = 0.03$) (Fig. 4). The angle of descent did not significantly affect the settling velocity of biodeposits ($p = 0.72$). Using the modified Stokes equation, mean pellet density was calculated to be $1.42 \pm 0.18$ g cm$^{-3}$ with a range of 1.13 to 2.26 g cm$^{-3}$.

To simplify calculations of horizontal particle transport, settling velocity was averaged across all particle sizes as $0.73 \pm 0.23$ cm s$^{-1}$. At this velocity, a particle would take ~5-10 min from release to deposition across the range of typical site depths (2–4 m). Oyster biodeposits had a mean tidal transport range of 72 m upstream at peak flood and 62 m downstream at peak ebb in the shallow (2m depth) section of the farm (Fig. 3B) and 118 m both upstream and downstream in the deep (4m depth) section of the farm (Fig. 3C).

## Erosion

Bottom shear velocities generated by the Gust chamber were in the range of those calculated from the bottom velocities measured by the ADV (Fig. 5A). Shear velocity from the covariance estimate ranged from 0.27 cm s$^{-1}$ during slack tide to 1.7 cm s$^{-1}$ at max ebb and flood. The lowest shear velocity tested in the erodibility experiments (0.95 cm s$^{-1}$) was within this range, and the higher shear velocities tested extended above the maximum field observations, with values up to 2.32 cm s$^{-1}$ (Fig. 5A, Fig. 6). There appeared to be more eroded mass at a given shear velocity under the farm than away and average total cumulative eroded mass under the farm ($6.82 \pm 2.52$ g m$^{-2}$) was over twice as high as that away from the farm ($3.22 \pm 0.34$ g m$^{-2}$). Additionally, eroded mass appeared to increase more with increasing shear velocity at the Farm than at the Away site, but only at the highest shear velocity did error bars not overlap between the two sites (Fig. 6).

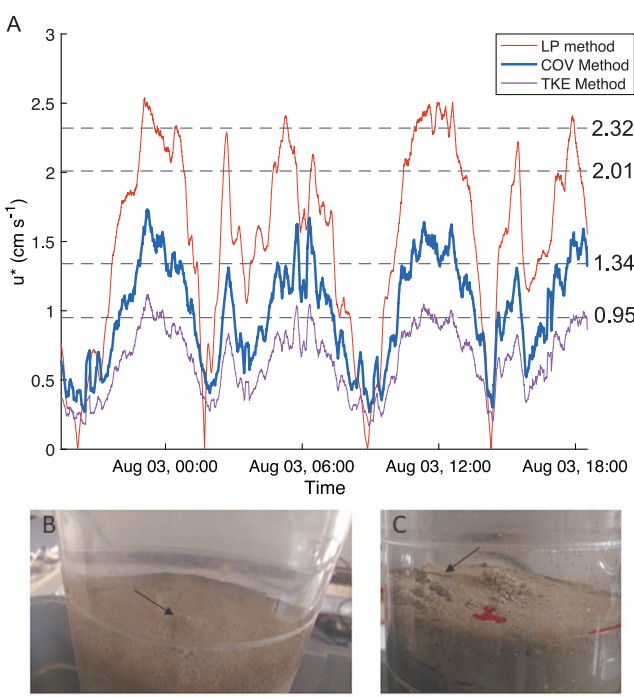

**Figure 5   Shear velocity and effects of shear on bottom.** (A) Time series of the bottom shear velocity (u*) at the Farm site calculated with three different methods; the logarithmic profile (LP), covariance (COV) and turbulent kinetic energy methods. The laboratory Gust chamber tested shear velocity levels marked with dashed lines. The horizontal line at $u^* = 2.01$ cm s$^{-1}$ marks the shear velocity level during which aggregates began collecting at the center of a core from the Farm site, shown in B (arrow). The line at $u^* = 2.32$ cm s$^{-1}$ marks the shear velocity during which partial mat failure was observed on the sediment surface in a core from the Away site, shown in C (arrow). Red marks are on the outside of the core. Photo credit: William C. Clemo.

There was visible evidence of erosion that was not quantified by the erosion chamber. Cohesive microbial mats were observed on the sediment surface in most erosion cores, and much of the eroded material was composed of larger particles or aggregates that were not entrained high enough in the erosion chamber to be taken up in the effluent and quantified by filtration. During the 2.01 cm s$^{-1}$ period of one Farm replicate, dislodged aggregates formed a mound in the center of the core (Fig. 5B). Partial mat failure was observed in the 2.32 cm s$^{-1}$ period of one Away replicate (Fig. 5C).

Longer-term data from the nearby LOBO buoy show that surface velocities under spring tidal conditions were nearly twice as high as during the period of the field sampling (Fig. 7). The scaling factor used to estimate bottom velocity from the relationship between the surface LOBO buoy and bottom ADV was 1.40 and showed a good fit ($R = 0.95$). Using the scaling factor, bottom velocities during spring tides were estimated as ~30 cm s$^{-1}$ during ebb and ~25 cm s$^{-1}$ during flood, with the maximum at ~40 cm s$^{-1}$. Neap tide bottom velocities were ~20 cm s$^{-1}$ during both ebb and flood, corresponding well with the bottom velocities measured by the ADV. During the spring tides, the calculated shear velocity approached three cm s$^{-1}$, higher than the highest shear velocities tested

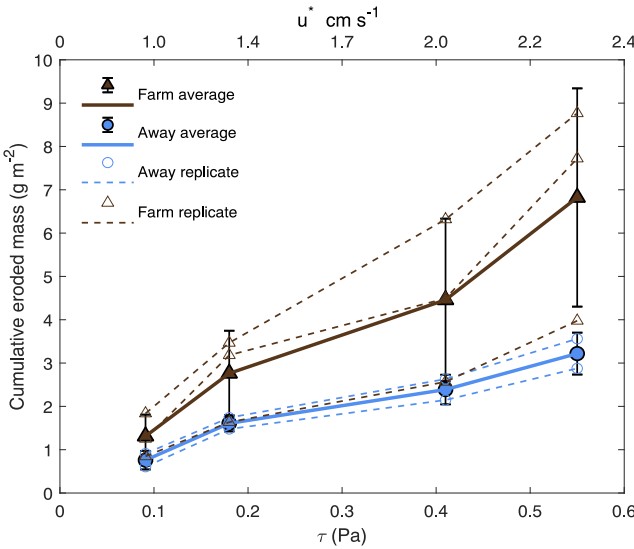

**Figure 6 Cumulative eroded mass.** Eroded mass for Farm (brown triangles; $n = 3$) and Away (blue circles; $n = 2$) sites at applied u* (shear velocity) and equivalent $\tau$ (shear stress). Smaller, unfilled symbols are replicates and larger, filled symbols are averages, and bars are $\pm$ one standard deviation.

in the erosion tests or encountered during the experiment (Fig. 5A). The long-term data also show that peak velocities were often associated with high turbidity events (Fig. 7). It should be noted that some of the high turbidity events were also associated with large precipitation events (triangles in Fig. 7), thus were likely driven by sediment runoff rather than resuspension.

## Geochemical characterization and biogeochemical fluxes

Sediments at both sites were poorly sorted coarse silt to fine sand (Fig. 8A, Table S1) (*Folk & Ward, 1957*). Organic content and water content were higher in muddier sediments as expected, and muddiness increased with depth at the Away site but varied with depth at the Farm site (Fig. 8, Table S1). The C:N (mmol g$^{-1}$ sediment) was 11.2 at the Away site and 11.48 at the Farm site (Table S1). Surface sediment chl-a measurements from Farm sediments were 14.3 and 11.8 µg chl-a g$^{-1}$ dry sediment and Away sediments were 7.3 and 22.5 µg chl-a g$^{-1}$ dry sediment. Phaeopigment concentrations were lower than chl-a in both samples from both the Farm site (5.88 and 6.80 µg g$^{-1}$ dry sediment) and Away site (11.05 and 8.74 µg g$^{-1}$ dry sediment).

Replication was not high enough to test if measurements of sediment fluxes differed between the two incubations, so data from both incubations were pooled by treatment (Away, Farm, and Away Amended, $n = 4$ for each treatment, Table S2). Sediment oxygen flux did not differ between the Farm and the Away sites, with negative values indicating flux into the sediment (Mann–Whitney Wilcoxon $p = 0.69$) (Fig. 9A). DIC fluxes out of the sediment did not differ among treatments (Kruskall–Wallis $p = 0.66$) but were highly variable, especially among the Away cores (Fig. 9B). NH$_4^+$ and PO$_4^{3-}$ fluxes were similar among all treatments (Kruskall–Wallis $p = 0.79$ and $0.31$, respectively) (Figs. 9C, 9D).

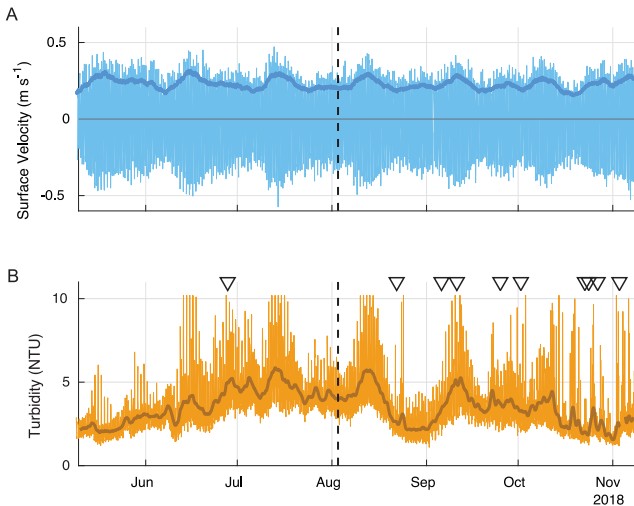

**Figure 7** **Surface velocity (A) and turbidity (B) during summer 2018.** The 40-hour velocity standard deviation (darker blue line) closely corresponds to the 40-h low pass filtered turbidity data (darker brown line). Some peaks in turbidity are associated with large precipitation events (triangles). The field survey time is shown with a dashed line. Positive velocity vales show landward flow and negative values show seaward flow.

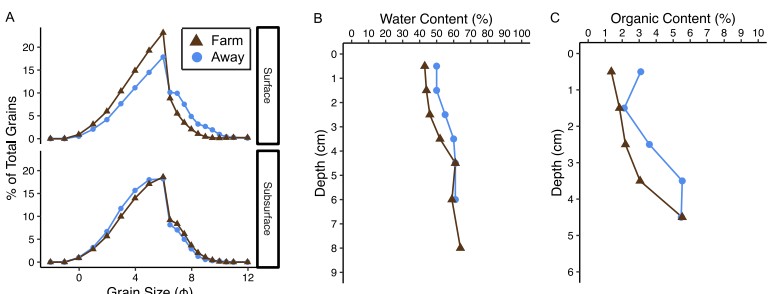

**Figure 8** **Sediment characteristics.** (A) Grain size distribution in percent for surface (0–3 cm) and subsurface (3–7 cm), (B) water content in percent, and (C) percent organic content at the away site (blue line and circles) and farm site (brown line and triangles).

There was no relationship between fluxes of $O_2$ and DIC ($p = 0.36$), $NH_4^+$ ($p = 0.44$) or $PO_4^{3-}$ ($p = 0.51$) (Figs. 9C, 9E, 9G).

A large nereid worm was found in a core taken from directly underneath the farm. This core also had the highest oxygen flux of all cores measured ($-72.5$ mmol m$^{-2}$ d$^{-1}$). A large nephtyid worm was found in an Away core that had the next highest oxygen flux measured ($-60.6$ mmol m$^{-2}$ d$^{-1}$). Removing these two cores from the sediment flux analyses did not result in significant differences among treatments (all $p > 0.1$) or significant relationships between nutrients and oxygen flux (all $p > 0.45$).

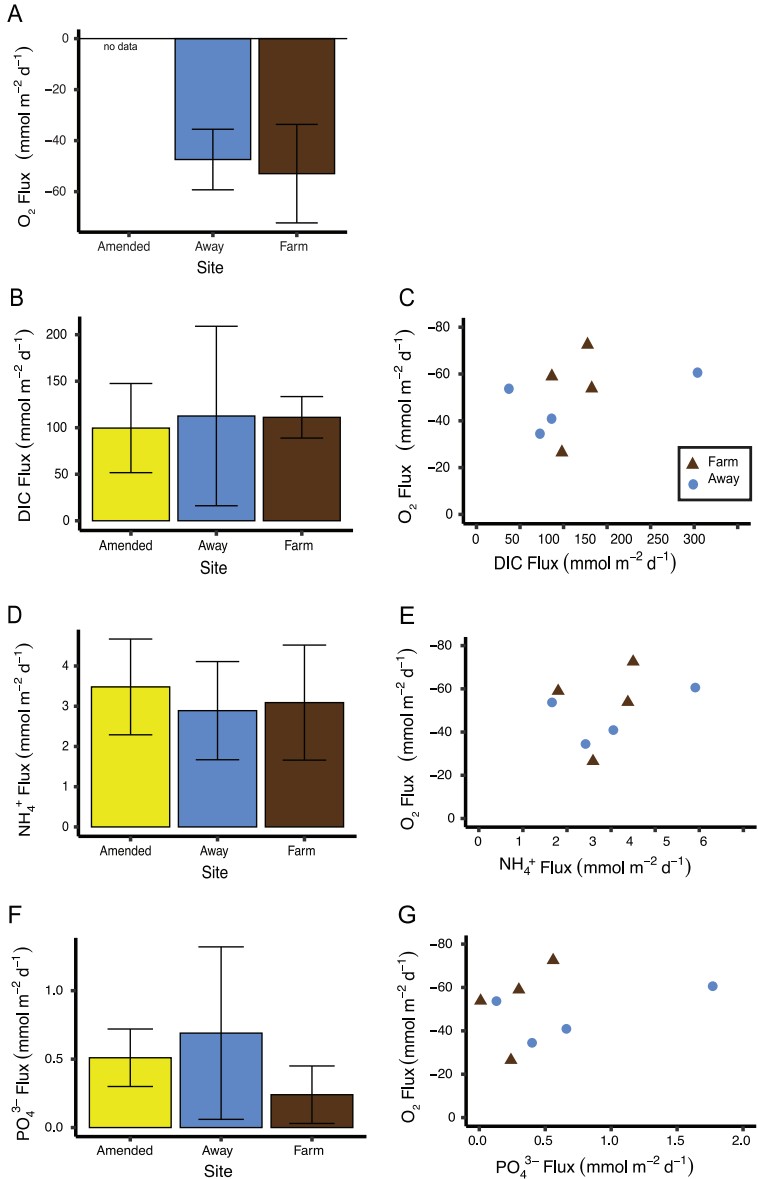

**Figure 9 Sediment biogeochemical fluxes.** (A) Oxygen, (B) DIC, (D) $NH_4^+$, and (F) $PO_4^{3-}$ fluxes. Positive value indicates flux out of the sediment. Also shown are relationships between oxygen flux and each nutrient flux (C, E and G). There was no significant relationship between oxygen flux and the DIC or nutrient fluxes. Because oxygen flux could not be measured from the amended cores, no data are shown for that treatment. Error bars are standard deviation.

## Macrofaunal community structure

Twelve families of macrofauna in 3 phyla were found among all samples collected (Fig. 10A, Table S3). Overall, macrofauna present at both sites were small-bodied and mainly consisted of burrowing, suspension-feeding bivalves (*e.g.*, Myidae, Mactridae), surface deposit-feeding polychaetes (*e.g.*, Spionidae, Flabelligeridae) and the burrowing, carnivorous polychaete, Nephtyidae. Abundance and taxa richness were highly variable

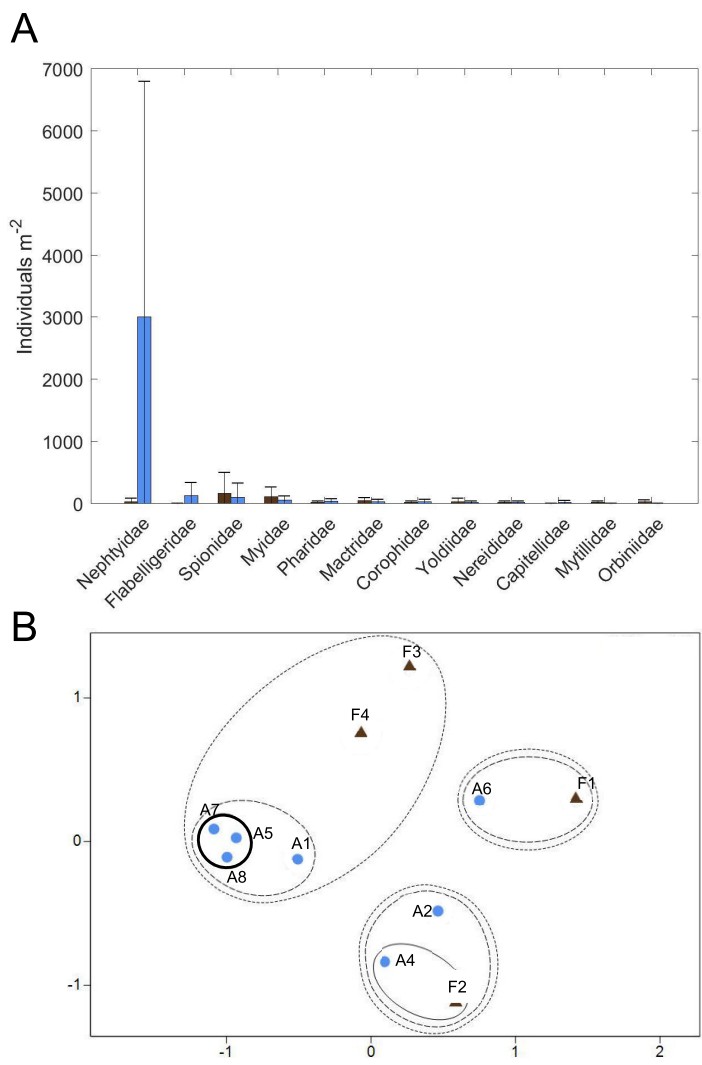

**Figure 10 Macrofaunal community structure.** (A) Average macrofaunal abundance by family for Farm (brown; $n = 4$) and Away (blue; $n = 8$) sites. Bars are ± one standard deviation. (B) Non-metric multi-dimensional scaling ordination of Farm (brown triangles; $n = 4$) and Away (blue circles; $n = 8$) replicate samples based on square-root-transformed macrofaunal abundances and Bray-Curtis similarities (stress = 0.11). Clusters at similarity levels of 20% (dotted line), 40% (dashed line) and 60% (solid line) are indicated. Bold solid line indicates Away replicates with high abundance of small nephtyids. Replicate 3 of the Away sites (Table S3) was excluded from the ordination because no macrofaunal animals were found.

at both sites. Family richness did not appear to differ between the sites, with 3.25 ± 1.89 families (mean ± st. dev.) at the Farm site and 4.0 ± 2.0 families at the Away site. Abundance appeared to be higher at the Away site, with 3,411 ± 3,902 individuals m$^{-2}$, compared to 459 ± 397 individuals m$^{-2}$ at the Farm site, but variability was high with the abundance of most families having a greater standard deviation than the mean (Fig. 10A, Table S3). Abundance was low in all macrofaunal groups except for small nephtyid polychaetes which dominated at the Away site, making up 88% of the abundance. Shannon diversity also

appeared similar but variable, with H' = 0.85 ± 0.22 at the Farm site and H' = 0.71 ± 0.60 at the Away site (Student's $t$-test, $p = 0.59$). Pielou's evenness (J') at the Farm site was 0.86 ± 0.18 and at the Away site was 0.54 ± 0.36 and was not significantly different between sites (Student's $t$-test, $p = 0.08$). There was no distinct partitioning in community structure between Farm and Away samples, although several of the Away site samples that shared relatively high abundance (> 6,000 m$^{-2}$) of nephtyids were > 60% similar (bold circle in Fig. 10B).

## DISCUSSION

### Farm footprint

The near-bottom velocities at the study site during the study period and lab measurements of biodeposit settlement rates were used to calculate a rough "footprint" of the impact of the farm on the sediment. An average-sized biodeposit released in the deeper section of the farm likely settled at a maximum of 118 m up or downstream of the farm at peak tidal flows, and far less if it was released in the shallower part of the farm (Fig. 3). At peak flow the smallest biodeposits in the deep section of the farm could have traveled up to 200 m from the farm before deposition, however transport of particles in the shallow section was more constrained (Fig. 3B), indicating that most particles would be settling within a far smaller range. Based on our footprint estimation, our selected Away site at ∼90 m downstream falls within the footprint of the farm and would therefore be expected to regularly but not continually receive oyster biodeposits.

There are few recorded measurements of oyster biodeposition settling velocities, however our mean rate of 0.73 cm s$^{-1}$ used in the "footprint" calculation is similar to previous measurements of ∼0.8cm s$^{-1}$ (*Haven & Morales-Alamo, 1972*) and, unlike previous investigations, was measured using biodeposits from the system of study. Our estimate of settling velocity is conservative, as it assumes biodeposits are sinking unaggregated. In situ, biodeposits may aggregate together, increasing settling velocity and reducing the farm's "footprint".

The weak correlation between settling velocity and surface area (Fig. 4) in addition to the non-zero intercept (we would expect particles with a surface area of 0 mm$^2$ to have a settling velocity of 0 cm s$^{-1}$) illustrates that either the relationship between surface area and settling velocity is nonlinear near zero, or that other parameters such as biodeposit density vary in a non-random way. We hypothesize that the scatter of data and the related poor fit likely results from differences in density among biodeposits, specifically that smaller biodeposits settle relatively faster because they are more dense than larger biodeposits. Oysters separate labile organic matter from refractory material and nonorganic material like sand and excrete these materials in feces and psuedofeces, respectively (*Haven & Morales-Alamo, 1966*). Biodeposits at the lower end of our range of densities (1.14 g cm$^{-3}$) are likely composed primarily of organic matter, while those at the upper end (2.26 g cm$^{-3}$) likely consist primarily of nonorganic minerals. Presumably, the relative makeup of these deposits depends on the composition of the suspended material in the water column at the time of filtration, affecting the densities of both feces and pseudofeces. We were unable to

differentiate between feces and pseudofeces in this study, and did not measure the density of individual pellets, preventing stronger correlations between biodeposit characteristics and settling velocity.

The biodeposits of different species of bivalves also appear to have different settling velocities, and larger bivalves tend to produce more biodeposits than smaller ones (*Haven & Morales-Alamo, 1972*). It is therefore important to consider the species and the size of bivalves when estimating the area of impact. For example, *Callier et al. (2006)* found that settling velocities for marine mussel biodeposits ranged from $\sim$0.3–1.8 cm s$^{-1}$ and varied based on the size of the individual. These higher settling velocities would result in an area of impact smaller than that calculated here based on oyster biodeposits, which may lead to greater rates of accumulation, particularly if individual bivalves were larger and produced more biodeposits. We did not account for individual size in this study, which could have been a source of variability in both the mean settling velocity and mean deposition rate.

Cage-area normalized rates of deposition ranged from roughly one third to one half of the deposition rates measured at the bottom mooring underneath the farm, suggesting that particle deposition beneath the farm consists of more than just oyster biodeposits. While all our deposition rates were within the range of other literature values, $\sim$10–650 g m$^{-2}$ d$^{-1}$ (*Comeau et al., 2014*), oyster biodeposition rates can vary widely with season (*Mitchell, 2006*; *Comeau et al., 2014*) and oyster size (*Haven & Morales-Alamo, 1972*), so direct comparisons between farms should be done with that in mind. Deposition at the Away site was particularly high, however in hindsight we suspect this may have been an artifact of the deployment method; the Away mooring was lowered to the bottom from the boat and was repositioned after deployment, which may have created a sediment plume that inflated settlement trap data. We also found that our Away site fell within the deposition footprint of the farm, leaving us unable to parse the relative impact of farm-sourced biodeposits and ambient particle deposition on total particle flux.

## Site comparison

The lack of difference in biogeochemical fluxes and macrofaunal community structure between the Farm and Away sites and the relatively low organic matter content indicate that biodeposition is not a major influence at either site, *i.e.*, that regardless of whether the Away site was within the footprint of the farm, deposition was small enough to have minimal impact. Previous studies in other systems have generally found significant differences in biogeochemical fluxes between farm and references sites, however reference sites in these studies tend to be >500 m away from farm sites (*Testa et al., 2015*; *Richard et al., 2007*; *Giles, Pilditch & Bell, 2006*), considerably further than the Away site in this study, which was selected at only $\sim$90 m from the edge of the farm lease to avoid the flow effects of the constriction in the river downstream. A similar reference site distance was selected by *Mallet, Carver & Landry (2006)* in a study of an oyster farm in New Brunswick, Canada. There, the researchers also found no difference between sediment geochemistry at reference and farm sites. Additionally, any potential difference in fluxes between our two sites may have been obscured by high within-site variability (Fig. 9). *Callier et al. (2009)* also found trends, but no significant differences, in biogeochemical fluxes between "unimpacted"

reference cores and cores with increasing densities of mussels suspended above them to simulate farm deposition, with high variability within their core treatments.

Like the variability in erosion, microalgal aggregations on the sediment surface may have contributed to variability in fluxes. Microphytobenthos have been shown in several studies to regulate biogeochemical fluxes across the sediment-water interface (*Reay, Gallagher & Simmons, 1995*; *Cerco & Seitzinger, 1997*), even when cores are kept in the dark (*Sundbäck & Graneli, 1988*). Indeed, chl-a measurements from the incubation cores (7.3–22.5 $\mu$g g$^{-1}$ dry sediment) were higher than the highest measured by *Watling et al. (2001)* (5.24 $\mu$g g$^{-1}$ dry sediment) in sediments from a deeper area of the Damariscotta. Chl-a concentrations were also higher than concentrations of phaeopigments (which indicate degraded chlorophyll often due to grazing), implying abundant living microphytobenthos (*e.g.*, *Bianchi, Dawson & Sawangwong, 1988*). Chl-a was highly variable between samples, suggesting patchiness that may have contributed to variation in fluxes.

Although there was not a correlation between oxygen flux and DIC (Fig. 9), the ratio of DIC flux to $O_2$ flux was ~2, higher than predicted from stoichiometry and observed in high-Arctic sediments (*Rysgaard et al., 1998*), but similar to other studies in sediments under normoxic conditions and lower than in hypoxic sediments (*Lehrter et al., 2012*; *An & Joye, 2001*). This is also in line with a previous study by *Newell, Cornwell & Owens (2002)* on the influence of biodeposition on sediments. They performed dark incubations of biodeposit-enriched sediment cores with a microalgal mat and found low oxygen influxes relative to DIC efflux, attributing the skewed ratio to the diversion of oxygen into oxidizing reduced compounds (*Newell, Cornwell & Owens, 2002*).

The "Amended" cores that were enriched with biodeposits prior to incubation showed no differences in nutrient fluxes from unenriched Away site cores, which suggests that these sediments were already saturated with organic matter. The amount of biodeposits added was comparable to the deposition at the farm site over a day (8 mL × 1.4 g mL$^{-1}$ density of biodeposits spread over the area of a 15 cm diam core is 634 g m$^{-2}$; assuming the biodeposits were loosely packed, this would be an overestimate, so the actual amount added is closer to the 490 g m$^{-2}$ d$^{-1}$ collected in sediment traps under the farm). However, it is important to note that, because of the incubation equipment malfunction, the oxygen flux measurements and nutrient sample collections for the second set of cores did not occur simultaneously, so conclusions about the relationship between oxygen and nutrient fluxes should be drawn with caution.

There was relatively low organic matter content at both sites, supporting the idea that biodeposition is minimal. Surprisingly, sediment at the Away site appeared to be slightly muddier and have higher organic content than the Farm site (Fig. 8). While we expected sediments at the Farm site to be finer and have higher organic content because of increased biodeposition, the potential addition of larger particles from eroding oyster shell to sediment directly beneath the oyster cages could coarsen the sediment and dilute its organic content with large and mostly inorganic shell fragments. However, it is difficult to assess site differences from these data given only one core was taken from each site.

The Away and Farm sites had similar infaunal communities, except for the abundant small nephtyids in some of the Away cores. It is likely that these were juveniles, and the

differences in abundance due to patchy recruitment. There is a general absence of recent infaunal community data from this area of the Damariscotta, which makes it difficult to assess from the literature whether the communities at and around the farm are comparable to other locations in the estuary. Additionally, macrofaunal abundance and community assemblage appear very patchy in these sediments and were probably not fully described by our sampling. Also, the cores were stored in a tank with sediment containing large macrofauna from an unrelated experiment, so it is possible that the large nereid and nephtyid worms found when samples were sorted had migrated and established in our cores and were not representative of the natural community at our sites. However, many large nereid jaws were observed in the bulk material while samples were being sorted, as well as many juvenile nephtyids counted, which suggests that the large macrofauna were likely present in the cores when they were collected.

Other than the abundant, small nephtyids at the Away site, the macrofaunal assemblages at both Farm and Away sites mainly consisted of discretely-motile, surface deposit/suspension feeding polychaetes (spionids and flabelligerids) and suspension-feeding infaunal bivalves (myids, mactrids, and pharids). Relative dominance of suspension feeders suggests that flow is slow enough to avoid sediment destabilization but fast enough to maintain sufficient suspended food particle flux (*e.g.*, *Widdows et al., 2004*). Additionally, surface-deposit feeding may be particularly advantageous where an algal biofilm offers a concentrated high-quality food source at the sediment surface (*Decho & Lopez, 1993*; *Montserrat et al., 2008*). Such mats, however, may disadvantage subsurface deposit feeders by impeding sediment-water exchange of oxygen and reduced compounds (*e.g.*, *Hansen & Kristensen, 1997*). Subsurface deposit feeders contributed less than 20% to Farm and Away total abundance, excluding nephtyid abundance. Though our sampling occurred during a relatively quiescent period of low flow and turbidity, the spring tide several weeks prior (Fig. 7) would have produced high flow rates and sediment erosion potentially stressful for infauna.

### Potential flushing of biodeposits

The partial mat failure during the erosion experiments suggests the possibility that these sediments are at least periodically eroded, decreasing the impact of biodeposits on sediments under the farm. The high variability in erosion at the higher shear velocity levels (Fig. 6) suggests sediment erosion occurs suddenly and at shear velocities exceeded only at the highest flow rates recorded by the ADV during the ~23 h deployment. This is likely because a microphytobenthic mat stabilizes surface sediments, resisting erosion until a point of critical failure; this is consistent with the high chl-a values measured in the incubation cores and with the divers' observations of patchiness in the mat cover of the sediment surface (Fig. 2). Though only observed in one of our replicate cores, the partial mat failure at the highest tested shear velocity (2.32 cm s$^{-1}$) falls within the broad range of shear velocities at which biofilms fail. *Walker & Grant (2009)* performed erosion experiments on sand and sandy mud sediments with patchy cyanobacterial mats beneath mussel long lines, observing mat failure at around 1.5 cm s$^{-1}$. *Grant & Gust (1987)*, however, examined erosion of purple sulfur bacteria and cyanobacteria biofilms on sand and found critical

friction velocity increased from around 1 cm s$^{-1}$ for clean sands to around 4 cm s$^{-1}$ on sands with biofilms.

Massive sediment erosion was not observed in our lab experiments, even as the shear velocities tested exceeded the maximum calculated from the measured *in situ* bottom flow (Fig. 5A). However, the *in situ* bottom flow was measured only for a short period during neap tide, and higher flow velocities during spring tides would substantially increase shear on the bottom because bed shear stress is proportional to the square of current velocity. Using data from the LOBO station, we predicted that bottom velocity during the time of greatest flow in the spring-neap cycle that summer reached ∼40 cm s$^{-1}$, driving bottom shear velocities close to 3 cm s$^{-1}$. This would be well above the shear velocity that resulted in partial mat failure in our erosion experiments. Erosion of surface sediment under the farm during maximum velocities of spring tides probably flushed the deposited sediment away from the farm. The periodic peaks in surface turbidity at spring tides observed in the long-term record are consistent with episodic erosion and flushing of biodeposits. Relatively high eroded mass at two of the three replicates from beneath the farm suggests some flushing of recent biodeposits may have also occurred during the neap tide in which our observations were made (Fig. 6). The poor sorting of sediments at both sites at surface and depth, however, suggests that bulk subsurface sediments do not undergo frequent erosion and deposition events (Fig. 8A, Table S1). Frequent erosion and redeposition tends to result in graded deposition due to gravity sorting and therefore well-sorted sediment profiles (*e.g.*, *Morton, 1988*). It therefore seems likely that erosion is limited to surface, recently deposited sediments and that these sediments are not redeposited in bulk around the farm.

## CONCLUSIONS

We estimate that, during neap tide in this system, the average oyster biodeposit released in the deep section of the farm may be tidally transported up to 118 m before reaching the benthos. Our away site at 90 m distance would therefore be expected to fall within the farm footprint. Though our site comparison indicated no difference between the away site and the farm site, the lack of sediment organic enrichment suggests that this may be because biodeposition is generally low and is not resulting in harmful impacts at either site.

Furthermore, the combined observations from the erosion experiments and consideration of maximum flow rates during other times in the tidal cycle suggest a tidally-driven erosion-deposition scenario; biodeposits accumulate and microalgal mats form on the sediment surface within the footprint during the low-energy tidal period. Then, as flow rates increase during spring tide, the bed experiences sudden failure as increased flow velocities produce bottom flow greater than the critical shear velocity and surface sediment detaches from the benthos in fragments, resulting in patchiness that contributes to highly variable fluxes. Periodic, sudden mass failure of sediment integrity would also disturb sessile macrofauna, which have more difficulty surviving physical disturbance than mobile taxa (*Brenchley, 1981*). Loss of microalgal biomass due to erosion may reduce the favorability of the sediment to settling macrofauna which generally prefer settling

on algal biofilm-covered sediment to bare sediment (*Pillay, Branch & Forbes, 2007*; *Van Colen et al., 2009*). The switching between deposition and erosion would cycle based on tidal flow, intermittently "clearing" patches of sediment under the farm and preventing excessive buildup of biodeposits. Particularly given the large range of flow velocities this area can experience, future investigations into this possible scenario should conduct sampling at several times throughout the spring-neap tidal cycle to get a full picture of how physical forcing affects aquaculture waste deposition and erosion. Additionally, more spatial sampling should be conducted to corroborate a calculated footprint with an actual footprint, and more comprehensively map the farm's area of influence and resolve uncertainty as to the impact of biodeposition.

## ACKNOWLEDGEMENTS

Our thanks to Bill Mook, Jeff Auger, and Meredith White at Mook Sea Farm for their help and the use of the Perkins Point farm as our study site, and the University of Maine's Darling Marine Center for accommodations and use of their facilities. Special thanks to Kathleen Thornton for DIC measurements, Larry Mayer and Lee Smee for helpful discussions, and our reviewers for their thoughtful suggestions.

### Funding

This project was sponsored and funded by the University of South Alabama Department of Marine Sciences and was the result of a graduate class project (MAS 583). The funders had no role in study design, data collection and analysis, decision to publish, or preparation of the manuscript.

### Grant Disclosures

The following grant information was disclosed by the authors:
University of South Alabama Department of Marine Sciences: MAS 583.

### Competing Interests

The authors declare there are no competing interests.

### Author Contributions

- Kara Gadeken, William C. Clemo, Will Ballentine, Steven L. Dykstra, Mai Fung and Alexis Hagemeyer conceived and designed the experiments, performed the experiments, analyzed the data, prepared figures and/or tables, authored or reviewed drafts of the paper, and approved the final draft.
- Kelly M. Dorgan and Brian Dzwonkowski conceived and designed the experiments, authored or reviewed drafts of the paper, mentoring students, and approved the final draft.

### Field Study Permissions

The following information was supplied relating to field study approvals (i.e., approving body and any reference numbers):

Verbal permission obtained from Bill Mook, owner/operator of Mook Sea Farm, Walpole, ME.

### Data Availability

The velocity and turbidity during our experiment (Fig. S1), formatted tables of geochemical characteristics (Table S1), sediment fluxes (Table S2), and infauna (Table S4), and raw data files for all data used in the experiment are available as Supplementary Files.

### Supplemental Information

Supplemental information for this article can be found online at http://dx.doi.org/10.7717/peerj.11862#supplemental-information.

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
