# Peer review of "Transport of biodeposits and benthic footprint around an oyster farm, Damariscotta Estuary, Maine"

_PeerJ, doi:10.7717/peerj.11862_

## Round 0.1 · original submission · Major Revisions

· Academic Editor

Major Revisions

I now have comments back from two expert referees who have evaluated your work. As you can see below, both found merit in the study, but each also had serious concerns about some aspects of the manuscript that will need to be addressed before the work could become publishable. In particular, both referees question whether the authors have adequately characterized the sediments captured in the study. Both referees point out alternative possibilities and question the assumption that all materials caught in all sediment traps were biodeposits that explain the differences in metabolism. These questions leave significant uncertainty in the primary finding about the extent of the biological footprint of the farm that will need to be addressed in revisions.

If you decide to undertake these suggested revisions, please ensure that all review comments are addressed in a rebuttal letter and any edits or clarifications mentioned in the letter are also inserted into the revised manuscript where appropriate. It is a common mistake to address reviewer questions in the rebuttal letter but not in the revised manuscript. If a reviewer raised a question then your readers will probably have the same question so you should ensure that the manuscript can stand alone without the rebuttal letter. Directions on how to prepare a rebuttal letter can be found at: https://peerj.com/benefits/academic-rebuttal-letters/

·

Basic reporting

This manuscript is well organized and generally well-written. There were no substantive issues with presentation of the data. I think they set up the problem quite well and incorporate a useful subset of the literature in this area. Figure are fine, data sharing is fine, they state and test their ideas/hypotheses.

Experimental design

This is indeed "original" research within the scope of the journal. The key question is whether one can define the "footprint" of oyster aquaculture effects. This is one of the key concepts in defining the effect of suspended bivalve aquaculture on the benthic environment, with most studies poorly identifying sites with minimal and maximal footprint signatures. For every study that shows large export (i.e. Testa), there are studies (Higgins) in areas with poor flushing that suggest poor sediment conditions under floats.

This study includes most of the elements one would incorporate into the program - flow, resuspension, biodeposit deposition, benthic fluxes; while I have some methodological questions, the study design is a model for this kind of work. Key limitations are 1) it is a "snapshot" with no temporal coverage and 2) benthic fluxes were conducted without an illuminated part of the incubation. Limitation 1 is a minimal issue, this amount of work is sufficient for a short contribution like this. The presence of benthic "mats" suggest that although the authors want to remove the confounding effect of benthic photosynthesis, it is indeed very important and may have a number of effects. However, such photosynthesis likely does not change the main conclusions. (I'll address this again later).

The authors fairly present their data, it's wonderful to see the spreadsheets, and it is very easy for a practitioner to follow what they have done. I'll address some minor/modest issues below, but the authors clearly meet the journal's standards.

Validity of the findings

The transport and fate of aquaculture biodeposits is a topic of great interest both for basic science reasons, but also it has great value in considering the "nutrient ecosystem services" rendered by such aquaculture. Benthic systems and material fluxes in aquaculture have considerable variability in this study and virtually all others, more replication likely could not strengthen or change the conclusions.

The overall rates of O2 or DIC fluxes are really quite high for a shallow water system, but it does not seem at all clear that biodeposits account for all of the metabolism. Systems with high benthic algal production also can have such high rates without the subsidy from aquaculture. The control may indeed have minimal aquaculture effects (as was hoped) and the biodeposits may be spread so widely as to not have an effect in the area within the aquaculture facility, much less beyond this. It would be good to consider a broader range of possibilities. Either way, the aquaculture facility does not appear to create problematic environmental conditions in this macrotidal ecosystem.

Additional comments

I deferred a more detailed discussion of the benthic flux work to this section; I'll discuss this here and add some (much) smaller line by line comments. I have more than a general idea of benthic fluxes in aquaculture, I participated in a Roger Newell study in the Damariscotta River focused on mussel rafts, but with a small side effort at the same sites this study used. The flow/resuspension/sedimentation work was limited to the mussels, but fluxes were carried out at the oyster site. Our dark fluxes inside and outside the Mook facility are quite similar to this study's. We also did denitrification and it was negligible. Our illuminated incubations were 4.6 x lower for NH4 flux and our O2 fluxes showed high rates of gross photosynthesis (3.4 mmol m-2 h-1), consistent with your mat observations. The sediments were autotrophic so your observations of resuspension and export are the only way to sustain such high rates. The main observations in this manuscript are not altered by our more detailed fluxes however. The comprehensive approach to the problem shown in this manuscript make it a valuable contribution.

Smaller comments:

Line 305-307 I would agree if the background respiration rates were very low, they are not.
Line 309-311 I don't think the algae are really confounding effects, they may increase respiration in the dark by providing more fuel.
Line 435 Is the C:N ratio mass or molar?
Line 436-440 Areal rates are chla are more useful when considering them relative to observed rates. I don't know if you can make that switch, but it minimizes your depth of collection as an issue.
Line 502 ingestion or filtration?
Line 526-530 I think you hedged this just enough....
Line 553-556 Low O2 relative to DIC occurs in systems with iron sulfide burial, methane production, and benthic algal photosynthesis. Newell at al. 2002 showed this with biodeposit addition studies with illumination. Newell, R. I. E., M. S. Owens and J. C. Cornwell, 2002. Influence of simulated bivalve biodeposition and microphytobenthos on sediment nitrogen dynamics. Limnology and Oceanography 47:1367-1369.

Overall, this is a nice study.

·

Basic reporting

The manuscript is well written, with good introduction and background sections, excellent referencing, and generally good, high quality figures. The raw data are supplied and accessible. A few comments under this heading:
1. In Figure 1, please provide a distance scale bar for panel C.
2. In figure 3, please indicate the velocity and distance sign conventions.
3. In Figure 6, the legend duplicates “Farm replicate” and “Away replicate” is missing
4. In Figure 7, please state the location and sign convention of the observations in the caption.
5. Please provide captions/descriptions with the raw data files, and please be sure to identify units for all quantities.

Experimental design

The paper fits well with the aims and scope of PeerJ. The research questions are well-defined and meaningful. The study was well-designed (given the geographical constraints), carried out professionally, and reported clearly and completely. There is one major omission in the experimental design and there are several minor clarifications/corrections/additions needed in the Methods, however:
6. The major omission is that the material in the sediment traps was not characterized or compared, other than by mass to determine apparent settling flux. There appears (?) to be an implicit assumption that all materials caught in all sediment traps were biodeposits, but this was not tested in any way for comparison between trap replicates, trap locations, or bottom sediments. This assumption seems reasonable for the traps immediately beneath the cages, but not the near-bottom traps. Simple measurements such as grain size distribution and organic content might have sufficient; additional measurements such as Chl a and nutrient content would have been even better. Combined with the sediment trap conceptual data interpretation error described below, this results in significant uncertainty in the primary finding about the extent of the farm “footprint”.
7. In eq 2 on line (l) 199, shouldn’t it be us(t) as well as h(t)?
8. The authors describe 3 different methods of estimating bottom shear velocity and state that all three methods produced similar results (l 225). Then in the results they choose one without sufficient justification and show that the 3 estimates differ by as much as a factor of 2.5 (Figure 5), which is as much as a factor of 6 difference in bottom shear stress. It would be much better to focus on the one method that is arguably most reliable and to state clear reasons for adopting it.
9. In the description of the erosion experiments, please state the source of the replacement water used and whether it contributed to any of the variability between experiments.
10. In l 280, “along-shelf” should be replaced by “axial” or “longitudinal”.
11. In l 287, how were the sediment samples treated before grain size distribution measurements?

Validity of the findings

The primary finding that the downstream out-of-farm location was within the footprint of the farm is not well-supported. It is a reasonable supposition, but there are in fact equally likely explanations of the data that cannot be ruled out, at least without additional analysis and/or interpretation. The authors have convincingly shown that the in-farm and downstream locations are quite similar geochemically, but there are several important ways in which the sites are slightly different and the reasons for their geochemical similarity are ambiguous.
12. There is a major conceptual error in comparing the sediment trap data below the cages to the sediment trap data just above the bottom. While it is likely that the cage traps are good estimates of the biodeposit flux below the cages, the cages occupy only a small fraction of the total surface area of the farm. To represent the true farm-area-averaged biodeposit settling flux, the cage trap estimates need to be adjusted to account for the empty space between the cages (e.g., as in Testa et al 2015). The near-bottom traps, however, are arguably reasonable estimates of the area averaged near-bottom settling flux because of particle dispersion. For example, suppose that the cages occupied 40% of the surface area of the farm. Then a below-cage trap estimate of 500 g/m2/d would translate to a farm-area-averaged estimate of 200 g/m2/d, considerably less than the near bottom trap flux measured at either location.
The higher near-bottom flux numbers might be due to resuspension or to another source of settling material. This is where characterization of the material caught in the different traps might help to identify and compare source(s). It is not valid to assume that all trapped particles are biodeposits (as in lines 375-380) except directly beneath the cages.
13. I do not agree with the settling velocity fit shown in Figure 4. It would have made much more sense to fit a line forced through (0,0). It is not reasonable that a biodeposit with 0 surface area would have a settling velocity other than 0. Indeed, without the 4 apparent outliers in the lower right of Figure 4, a linear fit forced through (0,0) might be a good description. Given an approximately constant particle aspect ratio, it is reasonable to expect an approximately linear fit between particle surface area and settling velocity. It is also reasonable to hypothesize that the scatter of the results might be due to variable particle densities, as the authors have done.
14. Again, the justification for choosing to use the covariance method for estimating shear velocity (lines 399-400 and Figure 5) is unconvincing and the wide difference between the different methods does not promote confidence. Can a better justification be given (e.g., comparing to the equivalent drag coefficient referenced to 1 m above bottom)? And once a decision is reached, there is no need to show all three estimates.
15. The statement on l 405-406 that there was only slightly more eroded mass under the farm than at the Away site understates the factor of two difference observed. One might expect that the bottom under the farm has more fresh biodeposits that are more easily eroded, and the erosion test results support that inference. Even if the Away site is in the footprint of the farm, it is on the outer fringe and not subject to the same rates of biodeposition as under the farm.
16. I would describe the accumulation of a mound of dislodged biodeposits in the center of the microcosm as evidence of bedload transport rather than evidence of erosion, but this may be just semantics.
17. Reference to the tidal velocity and turbidity record from the LOBO buoy is informative and its use to extrapolate to more erosive conditions under spring tide is quite reasonable. I think it would add significantly to the presentation to show a comparison between the LOBO velocity and turbidity to the ADV velocity over the period of ADV observations only, perhaps as an addition to Figure 3? I suspect this would also show that tidal resuspension at the LOBO site during neap tide is not absent, but certainly is weaker than during spring tide.
18. Figure 8 provides evidence that the Away site is a bit different from the Farm site, and in a direction not consistent with expectation for a site less impacted by biodeposition. The surface sediments at the Away site are clearly both finer and more organic than at the Farm site. This is counter to expectation as expressed in the Introduction (lines 85-87) and indicates either that expectations were wrong or that some other factor is promoting greater accumulation of organic fines at the Away site.
19. The lack of clear biological and geochemical flux differences between the Farm and Away sites does not necessarily mean that the Away site is within the footprint of the Farm. This is acknowledged by the authors on lines 528-530. However, on lines 635-636 of the Conclusions, this lack of inter-site difference is described as supporting the idea that the Away site is in the footprint of the farm, with no caveats.
I think it far more likely that the alternate possibility on lines 528-530 is a better explanation – “that biodeposition is not a major influence on flux rates, i.e., that the Away site was beyond the footprint of the farm, but the deposition was small enough to have minimal impact.” In fact, there were apparent inter-site differences in the sediment erosion experiments and in the surface sediment composition, but the erosion rates were very low and the fine, organic sediment fractions were very low at both sites. A stronger conclusion of this study is that there was no evidence of harmful impact due to biodeposition, either under the farm or at a similar site slightly downstream. The evidence supports the finding that biodeposition is not a dominant source of organic sediment loading in this energetic estuary under this farming intensity, but not that the Away site is reliably in the footprint of the Farm.

---

## Round 0.2 · Minor Revisions

· Academic Editor

Minor Revisions

I have now heard back from the most critical referee on your manuscript who concludes that your revision has satisfied their initial concerns. They recommend some last final edits to the manuscript prior to it being published. The suggestions are minor and some might argue stylistic, but clearly the nuance matters to some readers. Regardless, I expect these suggestions will be easy to address. I am therefore returning the text to you to make any final minor changes you wish to the manuscript before moving it forward into production.

Thank you for selecting PeerJ as an outlet for your work.

·

Basic reporting

The manuscript is well written, with good introduction and background sections, excellent referencing, and generally good, high quality figures. The raw data are supplied and accessible. All my previous suggestions have been addressed.

Experimental design

The paper fits well with the aims and scope of PeerJ. The research questions are well-defined and meaningful. The study was well-designed (given the geographical constraints), carried out professionally, and reported clearly and completely. All of my previous comments have been addressed.

Validity of the findings

All of my previous comments and suggestions have been satisfactorily addressed, with 2 minor exceptions:
1. The abstract still states that the authors found “similar erosion rates at the farm and away sites”, whereas the revised text acknowledges that erosion rates at the farm site tended to be about twice as high as at the away site. I suggest that the abstract text be modified slightly to say something like “found slightly higher erosion rates at the farm than at the away site.”
2. In addition, the text under the Flow and Deposition section of the results still uses the term “biodeposition” for sedimentation rates in the traps not immediately under the oyster floats, at both the farm and away sites. This should be changed to “deposition”, since the bio contribution is unknown.

---

## Round 0.3 · accepted · Accept

· Academic Editor

Accept

Thanks for making those last couple of revisions. I am happy to move your manuscript forward into production at this point.